

# Genetic diversity and molecular evolution of *Ornithogalum mosaic virus* based on the coat protein gene sequence

Fangluan Gao[1], Zhenguo Du[1], Jianguo Shen[2], Hongkai Yang[1] and Furong Liao[3]

[1] Fujian Key Laboratory of Plant Virology, Institute of Plant Virology, Fujian Agriculture and Forestry University, Fuzhou, Fujian, China
[2] Inspection and Quarantine Technology Center, Fujian Exit-Entry, Inspection and Quarantine Bureau, Fuzhou, Fujian, China
[3] Inspection and Quarantine Technology Center, Xiamen Exit-Entry Inspection and Quarantine Bureau, Xiamen, Fujian, China

## ABSTRACT

*Ornithogalum mosaic virus* (OrMV) has a wide host range and affects the production of a variety of ornamentals. In this study, the coat protein (CP) gene of OrMV was used to investigate the molecular mechanisms underlying the evolution of this virus. The 36 OrMV isolates fell into two groups which have significant subpopulation differentiation with an $F_{ST}$ value of 0.470. One isolate was identified as a recombinant and the other 35 recombination-free isolates could be divided into two major clades under different evolutionary constraints with $d$N/$d$S values of 0.055 and 0.028, respectively, indicating a role of purifying selection in the differentiation of OrMV. In addition, the results from analysis of molecular variance (AMOVA) indicated that the effect of host species on the genetic divergence of OrMV is greater than that of geography. Furthermore, OrMV isolates from the genera *Ornithogalum, Lachenalia* and *Diuri* tended to group together, indicating that OrMV diversification was maintained, in part, by host-driven adaptation.

## INTRODUCTION

RNA viruses, many of which threaten human health or agricultural safety, form measurably evolving populations as a result of their high mutation rate and short generation times. Molecular evolution studies are useful in understanding the molecular bases of the adaptation, geographical expansion, and process of emergence of RNA viruses, which are key to the design of management measures (*Lauring & Andino, 2010*; *Moya et al., 2000*).

*Ornithogalum mosaic virus* (OrMV) is one of the most important viral pathogens of floricultural crops, causing severe leaf symptoms as well as flower deformation of the affected plants (*Burger, Brand & Rybicki, 1990*). Under natural conditions, OrMV has a wide host range, infecting plants of the genera *Gladiolus*, *Iris*, *Ornithogalum* and *Diuris* (*Burger & Von Wechmar, 1989*; *Kaur et al., 2011*; *Wylie et al., 2013*). In addition, OrMV can infect saffron corms (*Crocus sativus*) as described in our previous report (*Liao et al., 2017*).

Corresponding authors
Jianguo Shen, shenjg_agri@163.com
Furong Liao, lfr005@163.com

OrMV was first detected in the United States in 1940 (*Smith & Brierley, 1944*). After that, OrMV has been reported in Netherlands (*Bouwen & Von der Vlugt, 1989*), France (*Grisoni et al., 2006*), South Africa (*Burger & Von Wechmar, 1989*), Israel (*Zeidan et al., 1998*), India (*Kaur et al., 2011*), South Korea (*Cho et al., 2016*), Japan (*Fuji et al., 2003*), New Zealand (*Wei, Pearson & Cohen, 2006*), Australia (*Wylie et al., 2013*) and China (*Chen et al., 2009*).

OrMV is a member of the genus *Potyvirus*, which includes more than 100 viral species. The biology and molecular biology of OrMV have not been studied intensively. However, it is known that, similar to some well-characterized potyviruses, OrMV has a single-stranded, positive-sense RNA genome, encoding a single polyprotein which is cleaved into 10 mature proteins by three virus-specific proteases (*King et al., 2011*). Additionally, a short polypeptide (PIPO) is expressed by a +2 nucleotide frame shifting from the P3 crison, resulting in a P3-PIPO fusion product dedicated to movement of the virus *in planta* (*Chung et al., 2008*; *Wei et al., 2010*). Although only five complete genomes of OrMV have been determined, CP sequences of 36 OrMV isolates are publically available from GenBank. In this study, CP sequences were used to investigate the genetic diversity of OrMV and investigate the evolutionary forces responsible for the diversity. Our results will improve understanding of viral genetic variation and adaptive evolution, which may be helpful in developing sustainable management strategies for control of OrMV.

## MATERIALS & METHODS

### Virus isolates and sequence alignment

CP gene sequences with known geographic locations and host origins were obtained from GenBank database using its Batch Entrez facility (Table S1). Multiple sequence alignments were performed with MUSCLE codon algorithm (*Edgar, 2004*) implemented in MEGA5 (*Tamura et al., 2011*).

### Phylogenetic network and recombination analyses

Two different approaches were used to investigate the occurrence of recombination events in CP sequences. First, the aligned CP gene sequences of 36 OrMV isolates were analysed using the Neighbor-Net method in SplitsTree 4.13.1 (*Huson, 1998*). In contrast to traditional bifurcating phylogenetic trees, SplitsTree constructs phylogenetic networks which allows for reticulation in the evolutionary relationships among taxa. Such reticulation could highlight the presence of recombination and if present, we followed up with a second analysis.

Second, sequences involved in the recombination and breakpoints were determined by using RDP4 suite (*Martin et al., 2015*), which incorporates the algorithms RDP, GENECONV, BOOTSCAN, MAXCHI, CHIMAERA, SISCAN, and 3SEQ. For each putative recombination breakpoint, a Bonferroni correction $P$-value (with a cutoff point at $P < 0.01$) was calculated. All isolates recognized were considered probable recombinants, supported by at least four different algorithms in RDP4 with an associated $P$-value of $< 1.0 \times 10^{-4}$. Simultaneously, the recombinants were further confirmed by GARD (*Kosakovsky Pond et al., 2006*) implemented in the Datamonkey web interface (*Delport et al., 2010*). The reliability of recombination breakpoints was evaluated using a KH test. To avoid false

identification, only recombination breakpoints supported both by RDP4 and GARD were considered to be recombinants.

## Genetic diversity and population subdivision

To investigate the genetic variation of the CP gene of OrMV, haplotype diversity ($H_d$) and nucleotide diversity ($\pi$) were calculated using DnaSP 5.0 (*Librado & Rozas, 2009*). Hudson's estimates of $K_{ST}$ and $S_{nn}$ were used to determine the presence of subdivision in populations (*Hudson, 2000*; *Hudson, Boos & Kaplan, 1992*). Genetic differentiation among populations was also evaluated by $F_{ST}$ using Arlequin 3.5 (*Excoffier & Lischer, 2010*). The ranges of differentiation and corresponding $F_{ST}$ values were as follows: a moderate degree of differentiation, 0.05 to 0.15; a large degree, 0.15 to 0.25; and a substantial degree, >0.25 (*Balloux & Lugon-Moulin, 2002*). In addition, analysis of molecular variance (AMOVA) was conducted using Arlequin 3.5 (*Excoffier & Lischer, 2010*), with counties and host species as grouping factors to test for the effects of country and host on the genetic diversity of OrMV. The statistical significance of $\varphi$-statistics was tested based on 1023 permutations (default).

## Phylogenetic analysis

After the potential recombinants were excluded, the phylogenetic relationships were reconstructed using the Maximum Likelihood (ML) approach implemented in MEGA5 (*Tamura et al., 2011*). For the ML analysis, substitution saturation was measured by Xia's test implemented in DAMBE 5.3.8 (*Xia, 2013*). The best-fitting of nucleotide substitution model was determined using MrModeltest (*Nylander, 2008*). ML analysis was performed under the GTR+$\Gamma_4$ model using the corrected Akaike Information Criterion and the robustness of the ML tree topology was assessed with 1,000 bootstrap replicates.

## Bayesian tip-association significance testing for the geographic and host species

To determine the potential geographic and host-origin effects on OrMV CP diversification, Bayesian Tip-association significance (BaTS) testing was performed in BEAST 2.4.6 (*Bouckaert et al., 2014*). Three statistics of phylogeny-trait association were computed: association index (*AI*), parsimony score (*PS*) and maximum monophyletic clade (*MC*) calculated from the posterior set of trees generated by BEAST 2.4.6 (*Bouckaert et al., 2014*). The statistical significance against the null distribution of trees was assessed by comparing it with the randomized trees generated from 10,000 reshufflings of tip characters. All *P*-values <0.05 from the three statistics, with low *AI* and *PS* scores and a high MC score, were considered significant, indicating a strong phylogeny-trait association.

## Test for natural selection

Two different types of analyses were performed to test for natural selection using the CODEML algorithm (*Yang, 2007*) implemented in EasyCodeML (https://www.github.io/bioeasy/EasyCodeml). Firstly, the branch model was used to identify CP genes with a null model assuming that the entire tree has been evolving at the same rate (one-ratio model) and an alternative model allowing foreground branch to evolve under a different

rate (two-ratio model). Multiple testing was corrected by applying the false discovery rate (FDR) method (*Storey & Tibshirani, 2003*) implemented in R. The CP gene of OrMV was considered as evolving with a significantly faster rate in foreground branch if the FDR-adjusted $P$-value less than 0.05 and a higher $\omega$ values ($\omega = dN/dS$, synonymous to non-synonymous substitution rates) in the foreground branch than the background branches. Secondly, the site model was used to identify nucleotide sites in the CP-coding region that were likely to be involved in OrMV evolution. For the site model, six codon substitution models described as M0, M1a, M2a, M3, M7, and M8, were investigated. The M1a model assumes two categories of sites ($\omega_0 < 1$, $\omega_1 = 1$), whereas the M2a model adds a third set of sites ($\omega_2 > 1$) to the M1a model. The M3 model, with three categories of sites, allows $\omega$ to vary among sites by defining a set number of discrete site categories, each with its own $\omega$ value. The M7 model partitions all the sites into ten different categories with $\omega < 1$ and fits a beta distribution to $\omega$. In the M8 model, an 11th category is added to the M7 model allowing $\omega$ values >1. For each nested model, the likelihood ratio test (LRT) was conducted by comparing twice the difference in log-likelihood values ($2\Delta LnL$) against a $x^2$-distribution, with degrees of freedom equal to the difference in the number of parameters between models. Only a $P$-value of 0.05 or less in the all LRTs was considered to be significant. Additionally, pairwise $dN/dS$ ratios were estimated using the yn00 program of PAML (*Yang & Nielsen, 2000*). Isolates that $dS > 2\times$ the mean $dS$ estimated from all isolates, as well as isolate pairs for which $dS$ estimates approached 0, were removed as advised by *Finseth, Bondra & Harrison (2014)*.

## RESULTS

### Recombination analyses

Recombination is an important source of genetic variability in viruses. To investigate the role of recombination in the evolution of OrMV, the split-decomposition network analysis with the CP gene sequences of 36 OrMV isolates was performed. A phylogenetic network showing reticulation was obtained (Fig. 1), indicating conflicting phylogenetic signals that are possibly attributed to recombination among viral genomes. The sequences were then checked for recombination using the RDP4 package (*Martin et al., 2015*). Four unique recombination events were detected by at least three independent methods implemented in the RDP suite (Table S2). However, only one isolate, Glad-8, was identified as a recombinant, with a breakpoint in the nucleotide 256, as confirmed by GARD analysis with a high level of confidence (both LHS and RHS $p$-values <0.01). The recombinant was excluded from the phylogenetic and selection analyses below.

### Genetic diversity and population subdivision

OrMV isolates could be divided into two subgroups reflecting two different origins of OrMV or representing two divergent OrMV populations (Fig. 1). The haplotype diversity for both subgroup 1 and subgroup 2 was 1.000, whereas the nucleotide diversity for these two subgroups was 0.106 and 0.017, respectively. Haplotype diversity and nucleotide diversity for all OrMV isolates were 1.000 and 0.156, respectively, indicating a high genetic diversity in OrMV populations and among subpopulations (Table S3A). Three independent

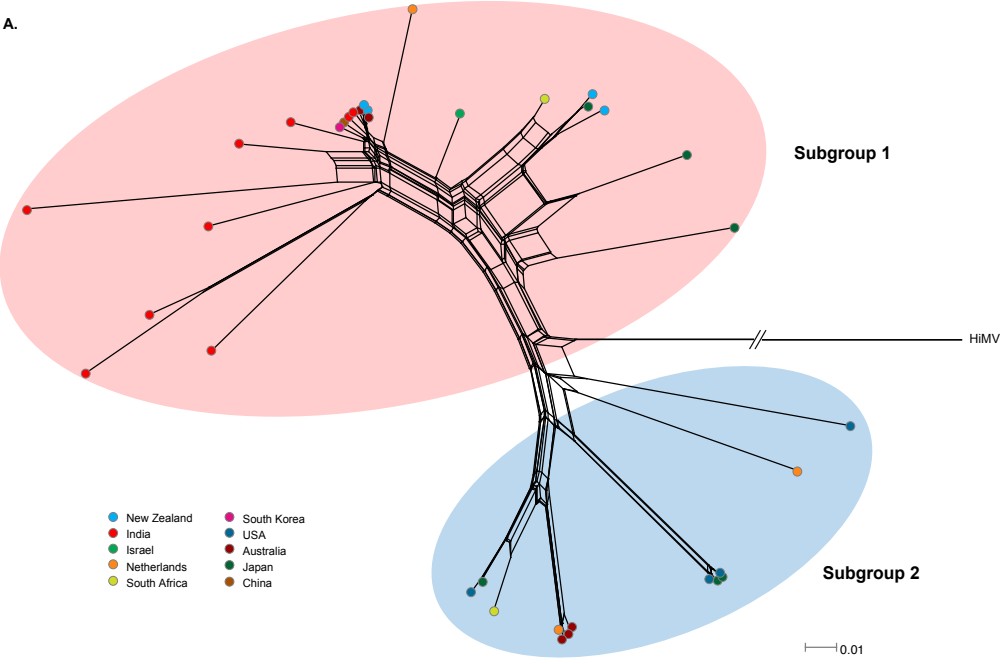

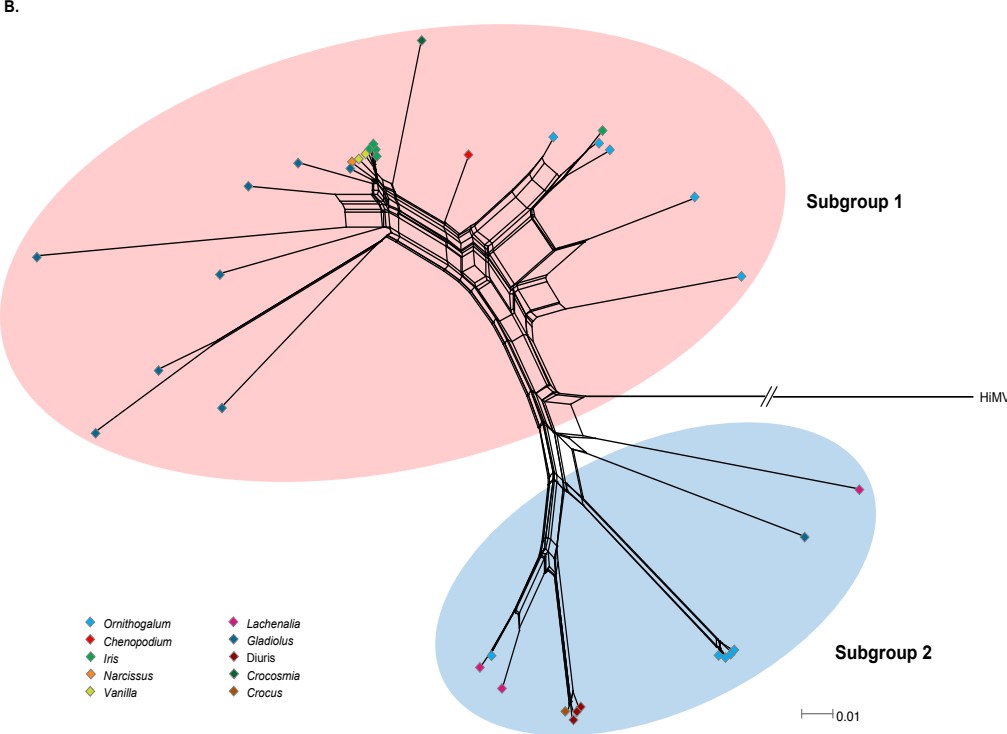

**Figure 1** Phylogenetic networks of the CP gene from 36 OrMV isolates from different countries (A) and hosts (B). *Hippeastrum mosaic virus* (NC_017967) served as an outgroup. OrMV isolates from different countries or hosts are indicated by a unique color. Branch lengths are proportional to the genetic distances.

**Table 1** Hierarchical analysis of molecular variance for the effects of geography and host species.

| Grouping factors | Source of variation | d.f. | Sum of squares | Variance components | Percentage of variation | Fixation index |
|---|---|---|---|---|---|---|
| Country | Among groups | 9 | 752.922 | 9.593 | 15.85 | $\Phi_{ST} = 0.159$[***] |
| | Within groups | 26 | 1323.717 | 50.912 | 84.15 | |
| | Total | 35 | 2076.639 | 60.505 | | |
| Host | Among groups | 9 | 939.994 | 18.492 | 29.73 | $\Phi_{ST} = 0.297$[***] |
| | Within groups | 26 | 1136.644 | 43.717 | 70.27 | |
| | Total | 35 | 2076.639 | 62.209 | | |

**Notes.**

Significance thresholds:

[***]$P < 0.001$.

tests of population differentiation were significant (Table S3B), indicating a great genetic differentiation between clade groups of OrMV.

To evaluate the role of geography and host specificity in shaping the population structure of OrMV, geographic regions and host genus were respectively used as a grouping factor to analyze the isolates of OrMV. When geographic regions were used as grouping factors, AMOVA tests revealed significant variation among geographic groups, making up 15.85% of the total variation, ($\Phi_{ST} = 0.159$, $P$- value <0.001) (Table 1). Similar results were obtained when host species was used as a grouping factor. Significant subpopulation differentiation was observed among groups ($\Phi_{ST} = 0.297$, $P$- value <0.001), which accounted for nearly 30% of the total variation of OrMV. Taken together, it seems that the effect of host species on the genetic variance of OrMV is greater than that of geography although both host species and geographic effects contributed to the genetic variance of OrMV.

## Phylogenetic analyses and BaTS testing

The ML phylogenetic trees based on the CP gene sequences showed that the 35 recombination-free OrMV isolates were grouped into two distinct clades with high bootstrap supports (Fig. 2A), consistent with the results of phylogenetic network analysis. With the exception of an isolate from Australia, no significant signal for geographic structure in the diversity of the CP gene was observed when the OrMV isolates were grouped by their geographic origins ($P_{MC} > 0.05$, Table 2). However, when the OrMV isolates were grouped by their host origins, a signal was found with more host-specific clustering than expected by chance, particularly for *Ornithogalum*, *Lachenalia* and *Diuris* ($P_{MC} < 0.05$, Table 2). The BaTS results indicated that OrMV CP diversification could be maintained in part by host-driven adaptation.

## Selection pressures

To investigate the differences in selective pressures behind the two clades (clade A and B) of OrMV, a two-ratio branch model test was performed using PAML, in which different $\omega$ values were assigned to the two clades. A LRT indicated that the one-ratio model should be rejected ($p < 0.05$, Table S4A); hence, selective pressures differed between the two clades. The mean $\omega$ values for clades A and B were 0.055 and 0.028 (Fig. 2A, Table S4A), respectively, indicating that clade B was subjected to stronger purifying selection than

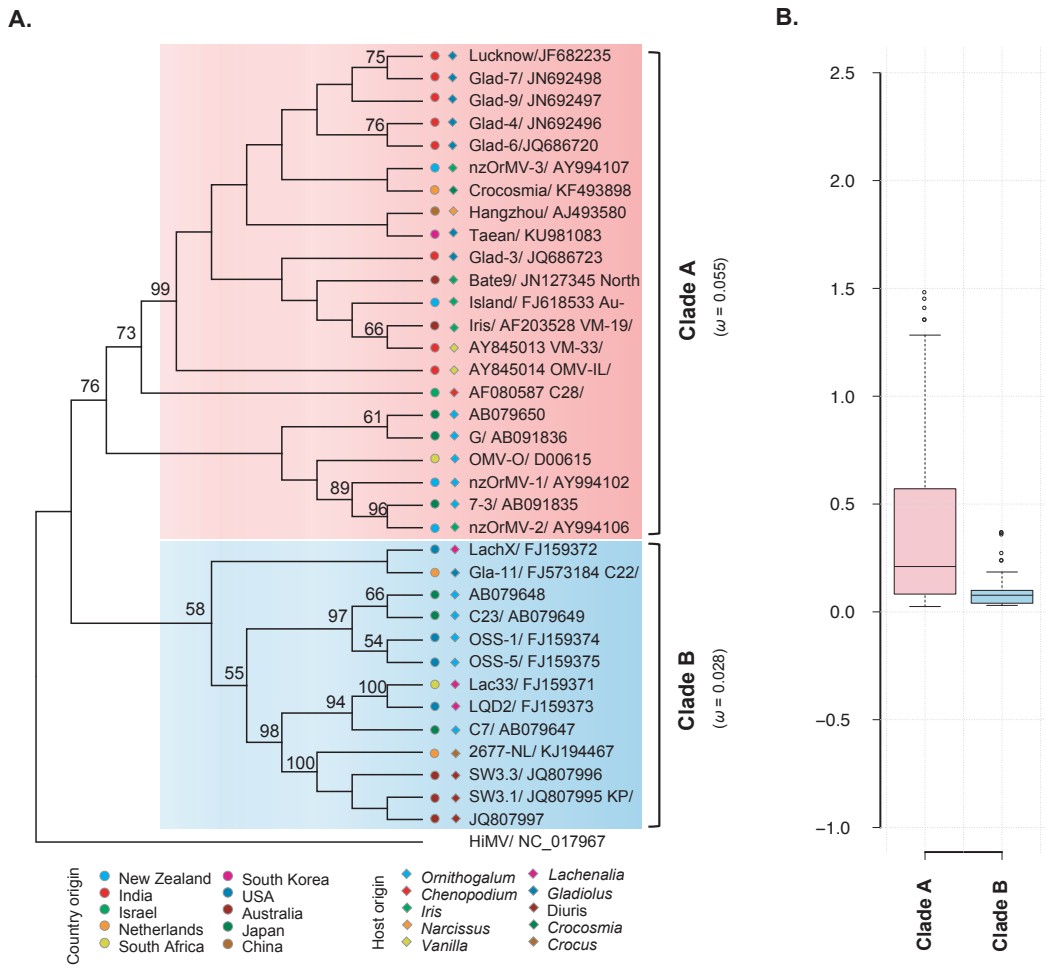

**Figure 2** **Evolutionary relationship of the CP gene from the 35 recombination-free OrMV isolates and comparison of *d*N/*d*S values between the two clades.** (A) ML phylogenetic tree showing genetic relationship among OrMV isolates. OrMV isolates from different regions (solid circle) and host species (diamond) are indicated by a unique color. Bootstrap percentage (BP ≥ 50%) are indicated above major branches. The distance unit is substitutions/site; (B) boxplots showing the *d*N/*d*S ratio of clade A (pink) and clade B (light blue) for the CP gene of OrMV.

clade A. Furthermore, the results from pair-wise analyses showed that there are differences between the distribution of *d*N/*d*S values between clade A and clade B (Fig. 2B). In the site model, there were no codons identified as being under positive selection and purifying selection was detected at the majority of polymorphic sites in the CP gene (Table S4B). Sliding-window analysis for sites under purifying selection was plotted in Fig. S1. Although the *d*N/*d*S values were below 1.00 for both clades, the *d*N/*d*S values of clade A were generally higher than those of clade B, indicating the CP gene in clade B had a stronger purifying selection pressure than those in clade A, in agreement with previous results from the branch model analysis.

**Table 2  Results of Bayesian Tip-association significance (BaTS) testing for the geographical and host species on the genetic diversity of OrMV.**

| Analyses | Statistic | $n$ | Observed mean (95% HPD) | Null mean (95% HPD) | $P$-value |
|---|---|---|---|---|---|
| Region | | | | | |
| | AI | | 1.920 (1.492, 2.314) | 3.094 (2.659, 3.464) | <0.001[***] |
| | PS | | 18.785 (18.000, 19.000) | 23.989 (22.051, 25.597) | <0.001[***] |
| | Asia | 17 | 2.235 (2.000, 3.000) | 2.485 (1.760, 4.002) | 0.860[ns] |
| | MC (Japan) | 6 | 1.987 (2.000, 2.000) | 1.281 (1.000, 2.000) | 0.080[ns] |
| | MC (Israel) | 1 | n/a | n/a | n/a |
| | MC (China) | 1 | n/a | n/a | n/a |
| | MC (India) | 8 | 2.224 (2.000, 3.000) | 1.485 (1.000, 2.129) | 0.180[ns] |
| | MC (South Korea) | 1 | n/a | n/a | n/a |
| | Oceania | 9 | 2.264 (1.000, 3.000) | 1.545 (1.003, 2.000) | 0.190[ns] |
| | MC (Australia) | 5 | 2.178 (1.000, 3.000) | 1.154 (1.000, 1.999) | 0.050[*] |
| | MC (New Zealand) | 4 | 1.012 (1.000, 1.000) | 1.114 (1.000, 1.767) | 1.000[ns] |
| | Africa | 2 | 1.000 (1.000, 1.000) | 1.014 (1.000, 1.021) | 1.000[ns] |
| | MC (South Africa) | 2 | 1.000 (1.000, 1.000) | 1.014 (1.000, 1.021) | 1.000[ns] |
| | North America | 4 | 1.999 (2.000, 2.000) | 1.158 (1.000, 2.000) | 0.090[ns] |
| | MC (USA) | 4 | 1.999 (2.000, 2.000) | 1.158 (1.000, 2.000) | 0.090[ns] |
| | Europe | 3 | 1.000 (1.000, 1.000) | 1.080 (1.000, 1.767) | 1.000[ns] |
| | MC (Netherlands) | 3 | 1.000 (1.000, 1.000) | 1.080 (1.000, 1.767) | 1.000[ns] |
| Host | | | | | |
| | AI | | 1.262 (0.896, 1.652) | 2.981 (2.451, 3.449) | <0.001[***] |
| | PS | | 13.007 (13.000, 13.000) | 22.136 (20.187, 23.975) | <0.001[***] |
| | MC (Ornithogalum) | 10 | 4.000 (4.000, 4.000) | 1.706 (1.001, 3.000) | 0.030[*] |
| | MC (Chenopodium) | 1 | n/a | n/a | n/a |
| | MC (Iris) | 5 | 1.493 (1.000, 2.000) | 1.224 (1.000, 1.996) | 1.000[ns] |
| | MC (Narcissus) | 1 | n/a | n/a | n/a |
| | MC (Vanilla) | 2 | 1.000 (1.000, 1.000) | 1.033 (1.000, 1.004) | 1.000[ns] |
| | MC (Lachenalia) | 3 | 2.000 (2.000, 2.000) | 1.046 (1.000, 1.028) | 0.020[*] |
| | MC (Gladiolus) | 8 | 2.224 (2.000, 3.000) | 1.451 (1.000, 2.244) | 0.200[ns] |
| | MC (Diuris) | 3 | 2.178 (1.000, 3.000) | 1.029 (1.000, 1.046) | 0.010[**] |
| | MC (Crocosmia) | 1 | n/a | n/a | n/a |
| | MC (Crocus) | 1 | n/a | n/a | n/a |

**Notes.**

AI, association index; PS, parsimony score; MC, maximum monophyletic clade; HPD, highest probability density interval; n/a, no data available because of insufficient sample size ($n < 2$).

Significance thresholds:

[*] $0.01 < p < .05$.
[**] $0.001 < p < 0.01$.
[***] $p < 0.001$.

## DISCUSSION

Recombination plays an important role in the evolutionary history of plant viruses, including potyviruses (*Moreno et al., 2004*) (*Gao et al., 2016a*; *Gao et al., 2017*; *Ohshima et al., 2007*), luteoviruses (*Pagán & Holmes, 2010*) and cucumoviruses (*Nouri et al., 2014*).

The greatest numbers of recombination events have been detected for the genus *Caulimovirus* (*Cauliflower mosaic virus*) in which the rate of recombination per base exceeds that of mutation (*Froissart et al., 2005*). However, the genetic variation generated by recombination is limited in OrMV and only one recombinant was observed in our analysis. There are two possible explanations. One is that the CP gene is a cold spot for recombination for OrMV. Such an idea has been proposed for some other plant viruses, such as *Chilli veinal mottle virus* (*Gao et al., 2016a*) and *Arabis mosaic virus* (ArMV)(*Gao et al., 2016b*). The other is that there is a strong selective pressure against the survival of OrMV recombinants. Consistently, purifying selection was detected at the majority of the polymorphic sites by two evolutionary analyses using the CODEML algorithm (Fig. S1), suggesting that most mutations in the OrMV CP gene were deleterious and consequently eliminated by natural selection.

Utilizing statistical models of variable $\omega$ ratios among sites, evidence of diversifying selection have been found in genes of potyvirus, such as *Potato virus Y* (*Moury et al., 2002*) and *Tobacco etch virus* (*Cuevas et al., 2015*). In this study, our results indicated that most codons of the OrMV CP gene were under purifying selection and no positively-selected amino acid site was identified. Strong selective constraint on the CP protein is probably attributed to the fact that it performs many different functions in the lifecycle of the virus, such as genome encapsidation, cell-to-cell movement, and plant-to-plant transmission (*King et al., 2011*). Interestingly, we found a difference in the selective constraints experienced by the two lineages of OrMV (Fig. 2, Table S4). In this case, the selective agents may be habitat differences between the two clades such as differences in the host species.

Geographic subdivision and host species contribute to the evolutionary dynamics of potyviruses, such as PVY, whose CP diversification was driven by both geographic and host-driven adaptations (*Cuevas et al., 2012*). In this study, the ML phylogenetic analysis did not show a clear geography-specific or host species specific clustering of OrMV possibly due to the occurrence of genetic exchange, but distinct genetic differences were discovered by AMOVA (Table 1) and BaTS (Table 2) analyses. The results of BaTS analyses provided evidence for host-specific clustering of OrMV isolates from the plant genera *Ornithogalum, Lachenalia* and *Diuri*, and we propose that to some extent, host-driven adaptation was responsible for the OrMV CP diversification. The possible role of geographic driven adaptation was not significant based on BaTs analyses at both large scale and finer scale. Therefore, geographic driven adaptation is not a major factor affecting OrMV CP diversification. Interestingly, a similar observation has been made for ArMV, a member of the genus *Nepovirus* of the subfamily *Comovirinae* within the family *Secoviridae* (*Gao et al., 2016b*). This suggests that OrMV and ArMV may share similar evolutionary mechanisms and that human activity has played a role in virus evolution because the introduction of ArMV and OrMV are more strictly controlled than that for PVY.

## CONCLUSIONS

In summary, this study represents the first attempt to understand the molecular evolution of OrMV. We found evidence of selective constraints in OrMV evolution and its diversification

was maintained partially by host-driven adaptation. However, isolates included in this analysis were relatively limited both in geography and host species. Further studies with larger, multiple-location and multiple-host-species sampling are needed to confirm our results and generalize the findings.

## ACKNOWLEDGEMENTS

We thank Dr. Sally Potter (Australian National University) and two reviewers for comments and suggestions about the manuscript.

### Funding

This work was supported by grants from the Natural Science Foundation of China (Grant No. 31570146), Fujian Natural Science Funds for Distinguished Young Scholar (Grant No. 2014J06008) and the Training Program of Fujian Excellent Talents in University. The funders had no role in study design, data collection and analysis, decision to publish, or preparation of the manuscript.

### Grant Disclosures

The following grant information was disclosed by the authors:
The Natural Science Foundation of China: 31570146.
Fujian Natural Science Funds: 2014J06008.
Training Program of Fujian Excellent Talents in University.

### Competing Interests

The authors declare there are no competing interests.

### Author Contributions

- Fangluan Gao conceived and designed the experiments, performed the experiments, analyzed the data, contributed reagents/materials/analysis tools, prepared figures and/or tables, authored or reviewed drafts of the paper, approved the final draft.
- Zhenguo Du authored or reviewed drafts of the paper, approved the final draft.
- Jianguo Shen conceived and designed the experiments, analyzed the data, authored or reviewed drafts of the paper, approved the final draft.
- Hongkai Yang prepared figures and/or tables, approved the final draft.
- Furong Liao conceived and designed the experiments, authored or reviewed drafts of the paper, approved the final draft.

### Data Availability

Sequence data in this study are available on GenBank under the accession numbers AB079647, AB079648, AB079649, AB079650, AB091835, AB091836, AF080587, AF203528, AJ493580, AY845013, AY845014, AY994102, AY994106, AY994107, D00615, FJ159371,

FJ159372, FJ159373, FJ159374, FJ159375, FJ573184, FJ618533, JF682235, JN127345, JN692496, JN692497, JN692498, JQ686720, JQ686722, JQ686723, JQ807995, JQ807996, JQ807997, JQ807998, KJ194467, and KU981083.

## Supplemental Information

Supplemental information for this article can be found online at http://dx.doi.org/10.7717/peerj.4550#supplemental-information.

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
