# Peer review of "Genetic diversity and molecular evolution of Ornithogalum mosaic virus based on the coat protein gene sequence"

_PeerJ, doi:10.7717/peerj.4550_

## Round 0.1 · original submission · Major Revisions

· Academic Editor

Major Revisions

I agree with the two reviewers that the Introduction needs more background information about the virus. Although use of English is correct please revise the MS for typos. Both reviewers have raised some issues about the statistical analyses in Methods and Results. As these analyses are key to this manuscript please answer and correct all points raised. Also I agree with reviewer#1 that since the number of sequences is low conclusions should be toned down. Please include suggested references.

Reviewer 1 ·

Basic reporting

The authors provide the first molecular evolution analysis for Ornithogalum mosaic virus (OrMV) using currently available sequences for the CP gene. For this, they perform phylogenetic, population structure and selection analyses.
I am not a native English speaker, but I think that language should be checked to correct several typos and/or sentences which are not clear to me. The structure of the manuscript is correct, although the introduction could be more informative; OrMV host range and geographic locations are provided, but information about other biological properties of the virus is not given, if known. This is particularly important, since the discussion also misses this information. The authors do not mention if raw data, i.e., sequence alignment, will be available in an open database or provided upon request.

Experimental design

The authors describe the methods employed in detail, and these are standard tools commonly used for molecular evolution analyses of plant viruses. The goals are clearly defined and analyses are refined and well conceived.

Validity of the findings

I have some main concerns regarding the validity of the results and the conclusions stated by the authors. Since the number of currently available sequences for this virus if still low, results and conclusions obtained with this dataset must be taken cautiously. For instance, the absence of sites under selection is considered a proof of strong purifying selection. Although this could be expected, available data are really scarce and authors should acknowledge. Also, the authors perform recombination analyses but, in my opinion, they should be more conservative when choosing non recombinant sequences. The fact that a short alignment is used in combination with a small number of sequences strongly recommends to remove all potential recombinants, i.e., those detected by at least three methods in RDP, from subsequent selection analyses. Finally, phylogenetics analyses do not provide a strong evidence of two different clusters; this grouping is not based on any biological property and thus phylogenetic classification should be taken cautiously again, since no other genomic regions are available to corroborate these preliminary results.
In conclusion, give the small sample size, most of the conclusions should be toned down.

Additional comments

Lines 103-105: I do not understand why the authors claim that the relaxed uncorrelated lognormal model is a better fit to the data than the strict clock model, but they still choose it to estimate the molecular clock.
Figure 1: Differences between figure 1A and 1B are not provided in the legend, neither in the main text. The same applies to Figure 2.
Table 1: Almost all data provided in this table is already provided in the main text, and then it is not necessary. It could be considered as supplementary table.
Figure 3: Sites under dN/dS=1 are indicated in orange, not “red”. This figure is not particularly informative and it would fit better as supplementary.
Table 3: A column indicating the sample size for each host/country is necessary. In fact, I guess that sample sizes will be particularly low in most of the cases, which would explain the lack of significance usually observed in MC cases. In this sense, for geographic analysis, and given the low number of samples, maybe it would be more reasonable to classify locations in areas wider than countries.

·

Basic reporting

It meets standards of English, article structure, but the background needs some work.

Experimental design

The research is original. It appears to be rigorous, but more details are required in the description of the methods. It is also unclear what the is knowledge gap, but this can be amended by improving the text.

Validity of the findings

Some statistical tests are not described in sufficient detail to assess the robustness of some of the results. Again, this can be improved with a better description of the methods.

Additional comments

In this study, Gao et al. conduct a range of population genetics and phylogenetic analyses to investigate the evolutionary dynamics of Ornithogalum mosaic virus. I value the effort that the authors have put into this study, but I think that it needs substantial additional work before it is published. I have suggested a range of statistical checks and analyses that the authors might have already done, but did not explain in the manuscript. Please see my specific points in the following sections.


Major points
- What criterium was used to choose the substitution model?

- What method for molecular clock model selection was used. Ideally, this would be marginal likelihoods via path sampling/stepping stone.

- The authors mention a 'lack of time stamp'. Does this refer to a lack of temporal structure? I assume that this was assessed via a root-to-tip regression or a date-randomisation test, but these are not shown. On that note, the authors calibrated the molecular clock using a prior distribution for the rate. To do this, it is important that the previous rate estimates corresponds to a very closely related virus and that it has strong temporal structure (i.e that it can be considered reasonably accurate) (see Rieux and Balloux 2016; Murray et al. 2016).

- An other important consideration of Bayesian phylogenetic analyses is that the influence of the prior should be assessed. This can be done by rerunning the analysis without data. If estimates of parameters of interest, such as the time to the most recent common ancestor, are reliable then their prior and posterior should differ substantially, such that the prior is not driving the inferences (Bromham et al. 2018; Ho and Duchene 2014).

- In the conclusion the authors acknowledge that their study has limited sampling of geography and host diversity. This can clearly bias inferences of trait evolution, such as geography or host range. This limitation needs to be discussed further.



Abstract
I think some of the language here needs to be explained better. For example, the authors mention BaTs analyses and estimates of \omega. These are measures phylogenetic signal and selective constraints, I suggest using these terms as opposed to just stating the program that was used.

Introduction
This section requires some more background information, such as why it is important to study the evolutionary dynamics of this virus, what is known so far, and what is the gap of knowledge that this study will address.

Minor comments
Line 36. Please not that not all viruses consist of measurably evolving populations. For example, many dsDNA viruses (including HPV, HBV, and Herpes) evolve too slowly to use sampling times of a few years as molecular clock calibrations.

Line 108. This part describes the selection of the tree prior. It is important to be consistent here are refer to the tree prior as a 'prior' rather than a model. More importantly, it states that the constant coalescent tree prior was chosen over the exponential growth, but there is no justification for this choice. This could be done via formal model testing, or by inspecting the posterior distribution of the growth rate. If this parameter includes zero, then one may resort to the constant size coalescent.



References

Rieux, A., & Balloux, F. (2016). Inferences from tip‐calibrated phylogenies: a review and a practical guide. Molecular ecology, 25(9), 1911-1924.

Murray, G. G., Wang, F., Harrison, E. M., Paterson, G. K., Mather, A. E., Harris, S. R., ... & Welch, J. J. (2016). The effect of genetic structure on molecular dating and tests for temporal signal. Methods in ecology and evolution, 7(1), 80-89.

Bromham, L., Duchêne, S., Hua, X., Ritchie, A. M., Duchêne, D. A., & Ho, S. Y. (2018). Bayesian molecular dating: opening up the black box. Biological Reviews.

Ho, S. Y., & Duchêne, S. (2014). Molecular‐clock methods for estimating evolutionary rates and timescales. Molecular ecology, 23(24), 5947-5965.

---

## Round 0.2 · accepted · Accept

· Academic Editor

Accept

Both reviewers agree that their concerns have been adequately addressed after extensively modification of the manuscript. However, the significance of the results is still a cause for concern. I think this will be improved when the sequence data is available. Nevertheless, I believe that your manuscript deserves publication presently.

Reviewer 1 ·

Basic reporting

no comment

Experimental design

no comment

Validity of the findings

no comment

Additional comments

I appreciate the modifications made by the authors to amend the minor and major weaknesses stressed in my previous review.

·

Basic reporting

OK

Experimental design

OK

Validity of the findings

OK

Additional comments

The authors have addressed my comments about Bayesian phylogenetics and other points that I raised.